# Role of Proteasomes in Inflammation

**DOI:** 10.3390/jcm10081783

**Published:** 2021-04-20

**Authors:** Carl Christoph Goetzke, Frédéric Ebstein, Tilmann Kallinich

**Affiliations:** 1Charité–Universitätsmedizin Berlin, Corporate Member of Freie Universität Berlin, Humboldt-Universität zu Berlin, Department of Pediatrics, Division of Pulmonology, Immunology and Critical Care Medicine, Augustenburger Platz 1, 13353 Berlin, Germany; 2Berlin Institute of Health at Charité–Universitätsmedizin Berlin, Charitéplatz 1, 10117 Berlin, Germany; 3Deutsches Rheumaforschungszentrum, Charitéplatz 1, 10117 Berlin, Germany; 4Universitätsmedizin Greifswald, Institute of Medical Biochemistry and Molecular Biology, 7475 Greifswald, Germany; ebsteinf@uni-greifswald.de; 5Charité–Universitätsmedizin Berlin, Corporate Member of Freie Universität Berlin and Humboldt Universität zu Berlin, Center for Chronically Sick Children, Augustenburger Platz 1, 13353 Berlin, Germany

**Keywords:** proteasome, inflammation, autoinflammation, autoimmune, proteasome-associated autoinflammatory syndrome

## Abstract

The ubiquitin–proteasome system (UPS) is involved in multiple cellular functions including the regulation of protein homeostasis, major histocompatibility (MHC) class I antigen processing, cell cycle proliferation and signaling. In humans, proteasome loss-of-function mutations result in autoinflammation dominated by a prominent type I interferon (IFN) gene signature. These genomic alterations typically cause the development of proteasome-associated autoinflammatory syndromes (PRAAS) by impairing proteasome activity and perturbing protein homeostasis. However, an abnormal increased proteasomal activity can also be found in other human inflammatory diseases. In this review, we cast a light on the different clinical aspects of proteasomal activity in human disease and summarize the currently studied therapeutic approaches.

## 1. The Ubiquitin–Proteasome System

The ubiquitin–proteasome system (UPS) is the most important intracellular non-lysosomal pathway for protein breakdown in eukaryotic cells [1,2,3]. As such, it plays a critical role in preserving protein homeostasis and protecting the cells from harmful protein aggregation which would compromise cell integrity and function [4,5].

The UPS is a highly complex mechanism with well over 1000 different genes involved [6], about only 50 of which encode proteins serving proteasome function [7]. The vast majority of the remaining UPS genes encode components involved in the selective modification of intracellular substrates with the ubiquitin molecule for recognition and subsequent degradation by proteasomes [8,9]. The covalent attachment of ubiquitin to target proteins is mediated by a cascade reaction involving three enzymes. In the first step, an E1 ubiquitin-activating enzyme uses energy released by adenosine triphosphate (ATP) hydrolysis to form a covalent bond between a cysteine residue of its active site and the C-terminal glycine of ubiquitin [10,11]. The activated ubiquitin is then transferred onto an E2 ubiquitin-conjugating enzyme, which itself can bind to one of several E3 ubiquitin ligases. In the final step, E3 ubiquitin ligases mediate the transfer of ubiquitin to a lysine residue of target substrates [12] (Figure 1). Less frequently, ubiquitylation occurs on other acceptors sites including cysteine, threonine and serine residues as well as N-terminal methionine [12,13,14,15]. Remarkably, ubiquitin itself may be also subjected to ubiquitylation at either one of its seven lysine (K) residues (K6, K11, K27, K29, K33, K48 and K63) or its N-terminal methionine (M1—known as linear polyubiquitylation), thereby triggering the formation of poly-ubiquitin chains [16,17]. Depending on the K-linkage used for ubiquitylation, these poly-ubiquitin chains may decide different possible fates for the attached protein [18]. Among these ubiquitylation types, K48-linked poly-ubiquitin chains represent canonical signals for proteasome-mediated degradation [17,19], while K63-linked and linear (M1) polyubiquitylation may support non-proteolytic roles such as signaling [19,20,21]. Importantly, the process of ubiquitylation is counteracted by deubiquitinating enzymes (DUBs) [22], whose most prominent families include the ubiquitin-specific proteases (USPs) and the otubain proteases (OTUs) [23].

As alluded earlier, K48-linked ubiquitination is a prerequisite for the breakdown of intracellular proteins by 26S proteasome complexes, which themselves are made up of a 20S core particle (CP) and a 19S regulatory particle (RP) [24,25]. The 20S CP is a barrel-shaped multicatalytic protease consisting of four heptameric αββα rings. The outer two rings are each composed of seven different α-subunits, while the inner two rings each contain seven different β-subunits [26,27]. The catalytic activity of the 20S CP is driven by the β1, β2 and β5 subunits encoded by the *PSMB6*, *PSMB7* and *PSMB5* genes, respectively [28]. All three β-subunits carry N-terminal threonine active sites exposed to the inner chamber of the 20 CP and exhibit chymotrypsin-, trypsin and caspase-like activities [29]. The assembly of the 20S CP is a highly coordinated process guided by the proteasome-assembly chaperones (PAC)1-4 encoded by the *PSMG1-4* genes and the proteasome maturation protein (POMP) encoded by *POMP* [7,30,31].

Importantly, 20S CPs are usually capped by regulators at either one or both sides of the barrel-shaped structure. One prime example of such regulators is the 19S RP [32], a complex of approximately 20 subunits which is essential for the recognition of K48-linked polyubiquitinated proteins via the Rpn10 and Rpn13 subunits [33,34]. The 19S RP also ensures the ATP-dependent unfolding of substrates as well as the removal of the ubiquitin moieties before translocation into the 20S CP [35,36,37]. Other regulators include the proteasome activators (PA) 28-αβ, PA28-γ and PA200 [27]. The ability of the 20S CP to bind distinct regulators at both sides gives rise to multiple proteasome complexes whose respective biological relevance, however, remains to be better understood [38] (Figure 1). Degradation products produced by the UPS are usually 8–10 amino acid long-peptides [39]. Only a small fraction of them may enter the endoplasmic reticulum (ER) via transporter associated with antigen processing (TAP) and are presented onto major histocompatibility (MHC) class I molecules [40,41]. In this regard, the UPS is a major contributor to MHC class I antigen presentation. Conversely, the vast majority of proteasomal products are further degraded into amino acids by various peptidases.

### 1.1. Proteasome Isoforms

Besides the β1, β2 and β5 subunits traditionally referred to as standard subunits, proteasomes may incorporate alternative catalytic β-subunits including the inducible β1i (low molecular weight protein 2—LMP2), β2i (multicatalytic endopeptidase complex subunit 1—MECL1) and β5i (LMP7) to form so-called immunoproteasomes (IPs) [42,43,44]. While standard proteasomes (SPs) are found in virtually all tissues, IPs are predominantly expressed in immune cells or other cell types that have been exposed to type I and/or II interferons (IFN) [43,45].

It is believed that IPs are more effective than SPs at degrading substrates under stress conditions [46,47,48,49], thereby protecting the cells from the accumulation of insoluble ubiquitin-modified protein aggregates [4,43,50]. It is also understood that SPs and IPs differ in their cleavage rates [51], thereby modulating the supply of MHC class I-restricted peptide positively or negatively, depending on antigen primary structure [6]. Recently, IPs have also been shown to regulate inflammation, as discussed below. Apart from the standard and inducible subunits, 20S CPs may contain the β5t catalytic subunit (encoded by the *PSMB11* gene) which is exclusively expressed in thymus [52]. The assembly of β5t results in the formation of so-called thymus-proteasomes which participate in T-cell positive selection [52,53]. Another proteasome isoform is the spermatoproteasome which carries the α4s (encoded by *PSMA8*) alternative structural subunit, predominantly found in testis and involved in spermatogenesis [54] (Figure 1).

### 1.2. Proteasomes and Cellular Pro-Inflammatory-Pathways

Due to its ability to degrade multiple regulatory proteins, the UPS coordinates a myriad of cellular responses. Innate immunity is a prime example of such processes, whereby the UPS regulates key signaling cascades. For instance, activation of the NF-κB and MAPK pathways in response to pattern recognition receptors’ (PRRs) engagement typically requires the generation of poly-ubiquitin chains at multiple levels. This is probably best depicted by the linear ubiquitin chain assembly complex (LUBAC) which contains the two E3 ubiquitin ligases HOIL-1L and HOIP [55]. By promoting the linear (M1-linked) ubiquitination of the IKK regulatory subunit NF-κB essential modulator (NEMO), LUBAC facilitates the phosphorylation of IκBα by IKK [55]. This phosphorylation serves as a signal for K48-linked poly-ubiquitylation and proteasomal degradation, thereby allowing the translocation of NF-κB into the nucleus for the transcription of genes encoding proinflammatory cytokines [56]. Further substrates of LUBAC include the receptor-interacting protein kinase 1 (RIPK1) involved in TNF signaling, which following linear ubiquitylation activates the proinflammatory NF-κB and MAPK pathways. The activation of RIPK1 is counterbalanced by the OTU DUB OTULIN, which is able to trim linear ubiquitin linkages [57]. In addition, K63-linked ubiquitin chains can result in the downstream activation of proinflammatory pathways, which is counteracted by the deubiquitinase A20 [58] (Figure 2).

Overall, proinflammatory pathways have been shown to be dependent on both proteasomal activity and ubiquitylation [6]. Toll-like receptor (TLR) stimuli, for example, rely on active IPs for the full induction of proinflammatory cytokines via the MAPK pathway [59,60]. These studies were mainly performed in myeloid-derived immune cells, but lymphoid cells were affected as well [61]. Interestingly, IP activity was found to play a critical role in T-cell differentiation with inducible subunits favoring T helper (Th)1 and Th17 differentiation, while SPs promoted a regulatory T cell phenotype [60,62,63]. B cells, especially plasma cells (PC), have an increased sensitivity to proteotoxic stress and therefore, are extremely dependent on proper proteasome function for their survival [64,65,66]. The accumulation of protein aggregates that cannot be effectively degraded by proteasomes generally induces intracellular stress and activates the unfolded protein response (UPR). This, in turn, leads to compensatory mechanisms including the upregulation of proteasome isoforms to restore protein homeostasis [6,67].

## 2. Impaired Proteasomal Function—(Mono)genetic Defects in the Ubiquitin Proteasome-System in Autoinflammatory Disorders

### 2.1. Proteasome-Associated-Autoinflammatory-Syndrome (PRAAS)

A cause-and-effect relationship between proteasome dysfunction and chronic inflammation was first established in 2010, as loss-of-function mutations in the *PSMB8* gene were identified in patients suffering from autoinflammatory syndromes [68]. As disease manifestations included joint contractures, muscle atrophy, microcytic anemia and panniculitis-induced lipodystrophy, these syndromes were initially referred to as JMP syndromes. Shortly afterwards, further mutations in the very same *PSMB8* gene were found in patients presenting with similar autoinflammatory symptoms. Many different names have been proposed to describe these disorders, including Nakajo-Nishimura syndrome (NNS) [69], Japanese autoinflammatory syndrome with lipodystrophy (JASL) [70] and chronic atypical neutrophilic dermatosis with lipodystrophy and elevated temperature (CANDLE) syndrome [71]. As these syndromes share the same genetic etiology, these were subsequently brought together and referred to as proteasome-associated-autoinflammatory-syndromes (PRAAS) [72,73].

Disease starts usually in infancy up to early childhood and is characterized by arthritis, skin eruptions, lipodystrophy and myositis as well as muscle atrophy in all PRAAS forms. Recurring fever is also described in all syndromes but JMP. Further PRAAS hallmarks include basal ganglia calcifications and hepatosplenomegaly [68,69,70,73,74,75]. As for the skin lesions, they seem to slightly differ in presentation between syndromes, with CANDLE being associated with annular plaque and violaceous eyelids [74], while NNS and JASL mostly present with nodular erythema [69,70]. Additionally, lesions in JMP patients are described as erythematous macular/papular and nodules [68].

Laboratory findings are in line with the observed chronic inflammation, as elevated C-reactive protein levels and erythrocyte sedimentation rate are common in patients [72]. Unfortunately, beside genetic analysis, rapid diagnostic tests for PRAAS are not available so far. However, two biological traits highly specific to PRAAS may be used to help establish the presence of the disease. These include: (i) a typical type I IFN gene signature in the blood [7,75,76,77] and (ii) a ubiquitous proteasome loss of function. One way to detect type I IFN responses is to monitor the expression of IFN-stimulated genes (ISG) at the transcript level by qPCR and/or Nanostring technology [78,79] or at the protein level (i.e., SIGLEC1) using flow cytometry [80]. Impaired proteasomal function can be measured either directly using activity-based-probes showing cleavage capacity [68] or indirectly by assessing the content of ubiquitin-modified proteins by Western blotting [69].

Given that the first PRAAS genetic mutations were all identified within the *PSMB8* gene encoding the catalytic IP subunit β5i [71], it was initially assumed that these disorders were primarily caused by IP defects. This notion was, however, rapidly challenged by the fact that PRAAS patients may carry genomic alterations in other proteasome genes such as *PSMA3* [76], *PSMB4* [76], *PSMB9* [76], *POMP* [76,81], *PSMG2* [82] and *PSMB10* [83]. Surprisingly, a series of proteasome loss-of-function mutations affecting *PSMB1* [84], *PSMD12* [85] or *PSMC3* [86] were not associated with typical PRAAS phenotypes, as they were found in patients suffering from neurodevelopmental delay (NDD). While cognitive impairment is also detectable in PRAAS patients, NDD subjects fail to develop any clinical signs of autoinflammation. The reasons for these discrepancies are unclear and warrant further investigation.

Due to the inherent type I IFN gene signature, PRAAS may be placed into the category of interferonopathies. However, in contrast to other well-defined interferonopathies such as Aicardi–Goutières syndrome (AGS) or STING-associated vasculopathy with onset in infancy (SAVI), the molecular mechanisms leading to type I IFN production in PRAAS remain unclear. One particularly attractive hypothesis for the induction of sterile inflammation in PRAAS subjects is the propagation of ER stress. It is indeed well established that ER associated protein degradation (ERAD) function is compromised by proteasome defects, thereby resulting in the retention of misfolded proteins in the ER [87,88]. Perturbed protein homeostasis in the ER lumen is then sensed by the three ER-resident transmembrane receptors ATF6 (activating transcription factor 6), IRE1 (inositol-requiring enzyme 1) and PERK (protein kinase R (PKR)-like endoplasmic reticulum kinase) which in turn initiate the so-called unfolded protein response (UPR). This results in the activation of downstream transcriptions factors destined to upregulate ERAD component and/or chaperones [89]. Strikingly, it has been shown that sustained UPR activation may induce inflammation even in a pathogen-free context by various mechanisms [67,90]. For instance, the exposure of microglia to proteasome inhibitors leads to the production of inflammatory cytokines in an IRE1-dependent fashion [91]. The observation that PRAAS patients express typical ER stress markers [81], suggests that the UPR might be one the mechanisms underlying inflammation in these patients. Interestingly, the activation of the UPR also results in the transcription of genes encoding inhibitors of the mTORC1 signaling pathway [92,93,94,95]. A decreased activation of mTORC1 would result in decreased lipid biosynthesis and reduced amounts of cholesterol. The observation that cholesterol deficiency is a danger signal alerting the innate immune system [96] reinforces the notion that the UPR might play a key role in PRAAS pathogenesis.

Thus far, the therapeutic options for PRAAS are extremely limited. Subjects with PRAAS respond poorly to conventional or biologic disease-modifying antirheumatic drugs [76]. Recently, major advances have been made by introducing the JAK1/2 inhibitor baricitinib into the treatment protocols [97]. Baricitinib, which blocks inter alia IFN signaling, has shown promising effects in PRAAS patients, as it could reduce disease manifestation in 8 out of 10 patients and promote clinical/inflammatory remission in 5 out of 10 patients, even though one patient had to drop out because of an uncontrolled BK-virus infection [97]. Similarly, successful treatment of PRAAS with tofacitinib, a pan-JAK-inhibitor, was demonstrated in a case report [98].

### 2.2. Further Genetic Inflammatory Diseases with a Link to the UPS

Dysregulated ubiquitylation has recently been associated with NF-κB-related autoinflammatory diseases, also named relopathies [99]. Among them, three are currently associated with altered ubiquitylation patterns [100]. These include mutations within genes encoding LUBAC and the OTU-deubiquitinase OTULIN and A20.

Mutations in *RBCK*, encoding HOIL-1L [101], and *RNF31*, encoding HOIP [102], are found in patients suffering from autoinflammation and immunodeficiency. These are traditionally referred to as LUBAC-mutations, as they result in the impaired assembly of the LUBAC complex, thereby preventing the linear (M1) ubiquitylation of critical signal transducing proteins (for TNF-signaling RIPK1, as depicted in Figure 2). LUBAC deficiency induces cell-type specific and cytokine-specific down- but also upregulation of the NF-κB-pathway. Clinically, patients with mutations in either one of the LUBAC subunits present with multiorgan autoinflammation, recurring infections and intracellular muscular glycogen inclusions with consecutive (cardio-)myopathy (amylopectionsis) [101,102].

OUTLIN, which counteracts linear LUBAC ubiquitylation, may be subjected to several loss-of-function mutations in patients exhibiting enhanced NF-κB activation [57]. Consequently, this leads to increased secretion of pro-inflammatory cytokines due to unbalanced LUBAC-activity (Figure 2).The resulting disease is named otulipenia or otulin-related autoinflammatory syndrome (ORAS) [103], whereby patients suffer from recurring fevers, sterile neutrophilia, lipodystrophy, panniculitis and systemic inflammation with growth retardation [100].

Another DUB involved in a negative feedback regulation is A20, whose alterations are associated with severe autoinflammation. Heterozygous loss-of-function mutations in the gene *TNFAIP3* encoding A20 result in a Behçet-like disease named A20 haploinsufficiency [104,105]. Autoinflammation manifests early and is accompanied by bipolar oral and genital ulcers, inflammation of the eyes, exanthemas, and arthralgia/arthritis (ref). Furthermore, mutations within *TNFAIP3* have been found in autoimmune–lymphoproliferative syndrome (ALPS) [105]. In these diseases, autoinflammation is typically accompanied by immunodeficiency due to the severe dysregulation of the NF-κB-pathway [100] (Figure 2).

USP18 deficiency was recently described as a novel autoinflammatory disorder [106]. USP18 serves as a negative feedback for type I IFN-signaling besides its role as a DUB for the ubiquitin-like modifier ISG15 [107]. Genetic mutations resulting in USP18 loss-of-function of present with a pseudo-TORCH (toxoplasmosis, other [syphilis, varicella, mumps, parvovirus and HIV], rubella, cytomegalovirus, and herpes simplex) fetopathy due to the unmitigated IFN-induced inflammation [106]. Similarly, impaired trafficking of USP18 to the IFN receptor results in loss of USP18 activity and a phenocopy of USP18 deficiency [108]. Untreated USP18 deficiency results in neonatal death [106]. However, immediate treatment with JAK-inhibitors has been successful in a first case [109].

Most recently, mutations within the major E1 ubiquitin-activating enzyme ubiquitin-like modifier activating enzyme 1 (UBA1) have been associated with adult-onset autoinflammation [110,111]. In the affected patients, somatic mutations in peripheral blood myeloid cells in the X-chromosomal *UBA1* gene resulted in the expression of a catalytically impaired UBA1 isoform. Mechanistically the impaired ubiquitin-activating activity resulted in decreased polyubiquitylation and increased unfolded protein response. Thus, only males were found to be affected and somatic nature explains the late onset of inflammation. Clinical phenotypes can vary. Common inflammatory manifestations include relapsing perichondritis, Sweet’s syndrome or vasculitis. Additionally, patients frequently present with hematological abnormalities including myelodysplastic syndrome. Due to the vacuoles found in myeloid precursor cells, the affected E1 enzyme on the X-chromosome, causing autoinflammation by a somatic the disease was named VEXAS [110].

## 3. Inflammation and Increased Proteasomal Activity

As the inducible β1i, β2i and β5i catalytic units are typically upregulated in response to inflammatory cytokines (as highlighted in the first chapter), the constitutive expression of IPs is a common hallmark of autoinflammatory diseases [112,113]. In this regard, other UPS-related biomarkers include proteasome-directed autoantibodies which were identified in multiple autoimmune diseases such as poly/dermatomyositis, systemic lupus erythematosus (SLE), Sjögren Syndrome (SjS) and multiple sclerosis (MS) [114,115,116]. Interestingly, these antibodies were capable of reducing proteasomal activation by PA28, at least in vitro [117]. Besides autoantibodies against proteasome subunits, circulating proteasomes were also found in various autoinflammatory diseases including SLE, rheumatoid arthritis (RA) or vasculitis [118]. Interestingly, the amounts of circulating proteasomes seemed to correlate with cellular damage, thereby making them good clinical markers for disease progression [119]. These findings led to multiple preclinical studies, investigating proteasomal activity in various autoinflammatory diseases and autoimmune diseases [59,60,120,121,122,123,124,125], as discussed below. Additionally, in autoimmune driven diseases further benefit from proteasome inhibition might be accounted to the specific proapoptotic effect on plasma cells [64,66,126,127].

### 3.1. Rheumatoid Arthritis

RA is a common autoimmune disease characterized by inflammation of the synovia and joints leading to cartilage and bone destruction [128]. It can be accompanied by systemic disorders, thereby potentially leading to high morbidity and increased mortality [128]. A key driver of RA pathogenesis is NF-κB [129], whose activation relies on proper proteasome function, as mentioned above. Hence, proteasome inhibition has proven clinically useful to reduce the production of interleukin (IL)-1β and IL-6 in T cells from RA patients [130]. In addition, blocking proteasome function by small molecule inhibitors has been shown to limit RA synovial cell proliferation by preventing the degradation of p53 [131]. Preclinical data from two mouse models of collagen antibody-induced arthritis and collagen-induced arthritis have also revealed that IP activity supports disease progression by promoting pro-inflammatory cytokine production and inflammatory joint infiltration [120].

### 3.2. Systemic Lupus Erythematosus

SLE is an autoimmune disease in which inflammation may target multiple organs, resulting in a very heterogeneous clinical manifestation with individual disease courses and various life-shortening and life-threatening complications [132]. Autoantibodies found in SLE target nuclear proteins are frequently produced by long-lived plasma cells (PC) that evade conventional B cell-targeted treatment [127]. However, due to their particularly high-rate production of secretory proteins, PCs generate large amounts of misfolded proteins in ER lumen that must be transported back to the cytosol by ERAD for subsequent degradation by proteasomes. For that reason, PCs are particularly sensitive to proteasome inhibition (as discussed below). A more direct involvement of proteasomes in SLE pathogenesis has been shown in mouse models in which specific IP inhibition was associated with reduced production of IFN-α [133], a critical disease marker [134]. Another hint for an active contribution of proteasomes to SLE comes from the observation that PA28γ is downregulated in lupus nephritis tissue [135]. Since PA28γ-capped proteasomes accelerate the turnover of phosphorylated STAT3 [135], it is highly likely that the decreased expression of PA28γ actively contributes to diseases pathogenesis. Interestingly, this notion seems to be specific to SLE, as the serum levels of circulating PA28γ in RA, SjS and other undifferentiated connective tissue diseases (CTDs) are increased and positively correlate with disease activity [136].

### 3.3. Sjögren Syndrome

Like RA and SLE, SjS is an autoimmune CTD. In SjS, autoimmunity leads to a chronic inflammation of salivatory and lacrimal glands [137]. It can manifest primarily or in combination with other autoimmune diseases and is the most frequent autoimmune CTD [137]. Besides the prevalence of anti-proteasome autoantibodies in SjS [138], the expression of the IP subunit LMP2 was significantly reduced in SjS salivatory glands [113,139]. The downregulation of LMP2 in SjS seems to occur as a consequence of increased protein turnover, as LMP2 mRNA were simultaneously upregulated. mRNA upregulation was most prominent in B cells and correlated with reduced susceptibility to proteasome inhibition [140]. However, proteasome inhibition in an animal model of SjS prevented disease development [141]. In this study, the beneficial effect of proteasome inhibitors was attributed to the prevention of Th17 differentiation and lymphocytic gland-infiltration. Furthermore, a variant of the *PSMB11* gene encoding the thymoproteasome specific β5t subunit was associated with the development SjS [142]. In this work, the introduction of this variant in mice resulted in impaired positive T-cell selection and an altered CD8+ T-cell receptor repertoire.

### 3.4. Inflammatory Bowel Disease

Besides CTD, it has been suspected that UPS dysfunction might be involved in the pathogenesis of other inflammatory diseases. For instance, it is thought that proteasomes participate in the progression of inflammatory bowel disease (IBD). The two main IBD forms are ulcerative colitis (UC) and Crohn’s disease (CD), both of which presents with relapsing chronic inflammation of the gut and sustained activation of the NF-κB pathway [143]. Strikingly, the IP subunits β1i and β2i were found to be constitutively expressed in colons of patients with CD [144,145] and to a lesser extent in UC [145]. It has been proposed that the presence of IP in patients with IBD favors IκBα degradation, thereby promoting the excessive activation of NF-κB activation [146]. IP subunit expression and activity in CD might be involved in immunopathogenesis. However, a prominent role of IFN-γ (known to induce IP formation) in CD limits conclusions made from these ex vivo studies [144]. Preclinical studies have been carried out to investigate whether such differential expression of proteasome isoforms was disease-relevant. It could be confirmed that both proteasome pan- and IP-specific inhibitions reduced gut inflammation in mice [147,148]. For the targeted approach, a dual inhibition of at least two IP subunits was necessary [149].The protective mechanism was attributed to reduced NF-κB-signaling in all these studies.

### 3.5. Multiple Sclerosis

MS is an inflammatory disease of the central nervous system (CNS) resulting in demyelination and neuronal damage [150]. It is the most common CNS autoimmune disease whose molecular pathogenesis remains, however, poorly understood. It predominantly affects young adults and has a severe impact on the quality of life of these individuals [150]. Interestingly, LMP2 and PA28αβ were found to be enriched in immune cells and oligodendrocytes of MS lesions [151]. Further evidence for an involvement of IPs in MS pathogenesis was made by the identification of a *PSMB9* variant associated with a reduced risk of developing MS. It is understood that the beneficial effect of this variant is attributed to altered MHC class I-restricted myelin-derived peptides [151]. It was also shown that proteasome pan-inhibition could prevent the development of experimental autoimmune encephalitis (EAE, a mouse model of MS) in mice following the injection of antigens and/or antigen-specific CD4+ or CD8+ T cells [125,152,153]. Similar effects were observed by specifically inhibiting the two IP subunits LMP2 and LMP7 [125,149].

### 3.6. Further Inflammatory Diseases with a Link to the UPS

The critical role of the UPS in cytokine signaling, especially the NF-κB- and Th17 differentiation pathways makes it a good disease-causing candidate in many other inflammatory disorders. For example, in psoriasis, a chronic inflammatory skin disease, driven by both innate and adaptive immunity, genome wide association studies have shown a direct cause-and-effect relationship between UPS dysfunction and psoriasis [154,155]. A central role here is assigned to the IL-23/Th-17 axis and TNFα a cytokine upregulated by NF-κB [156]. Similarly, any UPS dysfunction might trigger auto-immune diseases because of inappropriate supply of MHC class I-restricted peptides. This view is supported by a genome wide association study in Behçet’s disease [157].

## 4. The UPS as a Therapeutic Target

Bortezomib is the first proteasome inhibitor with FDA approval. It was initially described as an anti-inflammatory drug [158]. However, bortezomib and the second-generation proteasome inhibitor carfilzomib have only been clinically approved for treating malignancies such as multiple myeloma and mantle cell lymphoma so far [159,160,161]. Clinical data from a small study involving 12 patients with SLE [64] and a randomized double-blind controlled trial with 14 patients with SLE [162] have shown a beneficial effect of bortezomib on disease outcome. Similarly, in another study, bortezomib was shown to deplete PCs and reduce autoantibody production [127,163]. As discussed earlier, this effect can be easily explained by the fact that antigen-producing PCs need a high proteasomal capacity to degrade misfolded antibodies, which renders them susceptible to proteasome inhibition [127]. The perceived benefit of bortezomib is its proapoptotic effect on long-lived PCs. This effect is unfortunately not limited to pathogenic PCs but also depletes protective PCs [164]. Bortezomib has furthermore successfully been used in different cases of autoimmunity including autoimmune cytopenia, refractory primary SjS and encephalitis [165,166,167,168]. As all beneficial effects of bortezomib on the SLE disease course cannot be attributed to the reduction in autoantibodies [162], an additional anti-inflammatory effect is suspected. For instance, ex vitro stimulated T cells from healthy donors showed the reduced expression of inflammatory cytokines under proteasome inhibition by bortezomib [169]. A substantial part of the anti-inflammatory potential might also be attributed to overall cytotoxic effects on immune cells [170], which were most prominently observed in monocytes [124,170]. Similar effects on antibody formation has been observed in a small case series of anti-NMDA-receptor encephalitis [171], and a multicenter randomized controlled double-blinded study is currently recruiting patients [172] (ClinicalTrials identifier NCT03993262, https://clinicaltrials.gov/ct2/show/NCT03993262, accessed on 20 February 2021). However, a therapy with bortezomib is limited by the hematological and neurotoxic side effects [173]. The assumed effects are summarized in Figure 3.

Further proteasome inhibitors, particularly those only targeting the IP subunits, are currently under investigation [174], as they are thought to have fewer toxic side effects [175,176]. IP-specific inhibitors have been established in preclinical research [120,174] and have shown promising potential in multiple inflammatory disease models of various inflammatory and autoimmune disorders including experimental arthritis [120], sepsis-models [59], experimental autoimmune myocarditis [60] experimental autoimmune encephalomyelitis [125] and experimental colitis [121] inter alia [123,177,178,179].

For sufficient anti-inflammatory treatment, a co-inhibition of at least two IP catalytic subunits is necessary [149,179]. For immune cell depletion, including PC depletion, at least a partial added inhibitory effect on SPs is needed, as the catalytically active SP subunits β1, β2 and β5 are upregulated compensatory under highly specific IP inhibition in these cells [180]. The first human immunoproteasome inhibitor with such inhibitory capacity is KZR-616 [174]. It is currently in clinical phase 2 studies for treating systemic lupus erythematosus (ClinicalTrials identifier NCT03393013, https://www.clinicaltrials.gov/ct2/show/NCT03393013, accessed on 20 February 2021) and polymyositis/dermatomyositis (ClinicalTrials identifier NCT04033926, https://www.clinicaltrials.gov/ct2/show/NCT04033926, accessed on 20 February 2021).

Whilst proteasome inhibitors are well established, only limited data are currently available on the therapeutic potential of small molecule proteasome activators [181]. Examples of the very few substances capable of activating proteasomes include a MAPK inhibitor [181] and the protein kinase (PKA) activator Rolipram, which acts most likely by phosphorylating proteasomes [182]. Other substances increase proteasomal activity by the transcription of proteasome genes [183] or by the inhibition of DUBs [184] Clearly, further research is needed in this field, as increasing proteasome activity may provide treatment for patients with PRAAS or other diseases [181].

## 5. Conclusions

To summarize, the UPS is tightly involved in inflammation and (auto-)inflammatory diseases. It is involved in regulating pro-inflammatory pathways and is in turn upregulated during inflammation. Its exact role in the disease pathogenesis of inflammatory diseases is, however, still under investigation. Then again, impaired proteasomal activity is known to induce sterile inflammation, as observed in PRAAS patients. The exact mechanism is still not fully understood. Unravelling the pathogenesis of such diseases further will aid in better therapeutic approaches for PRAAS patients. For (auto-)inflammatory and auto-immune diseases, multiple mechanisms involved have been found. This resulted in successful initial clinical trials using the immunoproteasome as a therapeutic target. Larger studies with novel proteasome inhibitors with an optimized side-effect spectrum seem promising for larger randomized clinical trials.

## Figures and Tables

**Figure 1 jcm-10-01783-f001:**
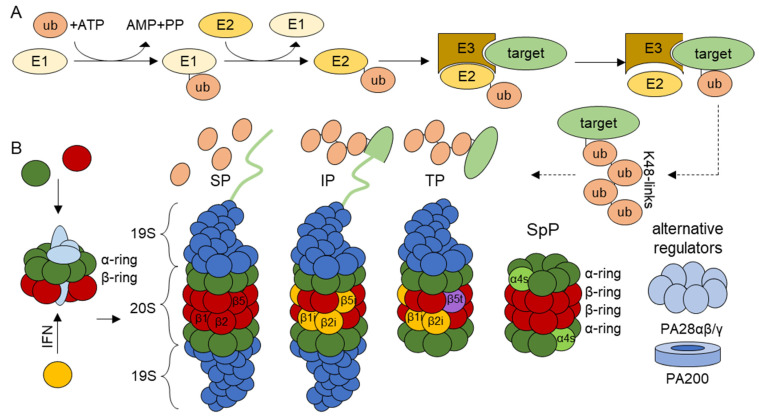
**Schematic representation of ubiquitylation and proteasomal protein degradation.** (**A**) Proteins destined for proteasomal degradation are conjugated with ubiquitin in a three-step cascade. First, E1-ubiquitin-activating enzymes bind to ubiquitin in an ATP-dependent reaction. This ubiquitin is transferred onto a E2-ubiquitin conjugating-enzyme. The E3-ubiquitin-ligase binds to both ubiquitin-conjugated E2-enzymes and target substrates which thereby undergo modification with ubiquitin. Ubiquitylated proteins can be polyubiquitylated. Depicted is a K48 linked polyubiquitylation, where ubiquitin is consecutively attached to the lysine 48 of the already bound ubiquitin. (**B**) The 20S proteasome core particle is made up of α- and β-subunits. The assembly to αββα asymmetric heptameric rings is guided by assembly chaperones. Each 20S core particle consists of two outer α-rings and two inner β-rings. In standard proteasomes (SPs), the catalytically active subunits are β1, β2 and β5. In immunoproteasomes (IPs), these subunits are replaced by the inducible subunits LMP2 (β1i), MECL1 (β2i) and LMP7 (β5i). IPs are preferentially incorporated to newly synthetized proteasomes in response to IFN, as indicated. Additional isotypes include the thymoproteasome (TP) which contains a unique β5t protease subunit and the spermatoproteasome (SpP) that incorporates a specific structural α4s subunit. The active sites of the catalytic subunits face the inside of the 20S barrel shape. Proteasomes can bind to different regulators on one or both sides. The 19S regulator has receptors for poly-ubiquitylated proteins and helps to unfold the proteins, remove ubiquitin from substrates and translocate them into the 20S for degradation. The regulators can attach to different isoforms on either or both sides. Additionally, combinations of the 19S regulator and PA28αβ/γ or PA200 regulators exist.

**Figure 2 jcm-10-01783-f002:**
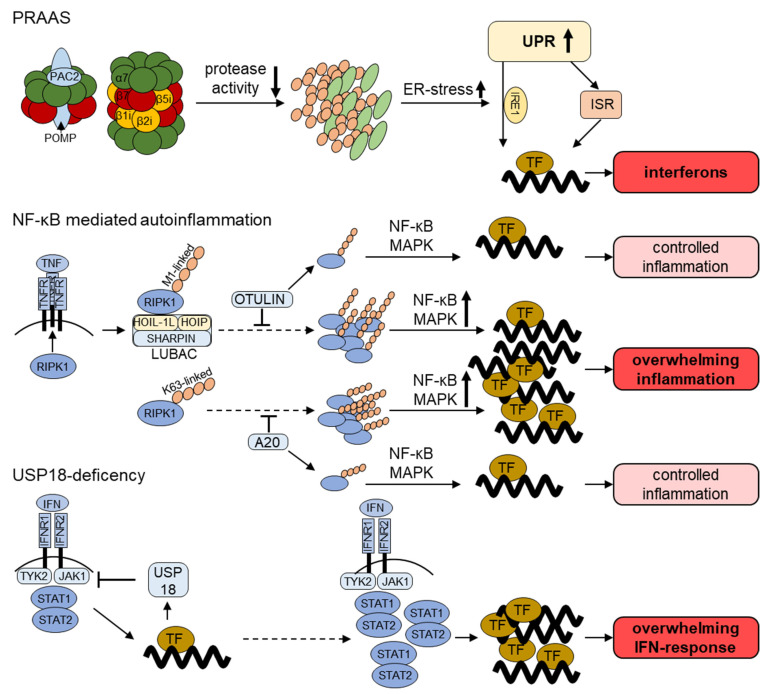
**Current understanding of the pathogenesis of UPS dysfunction in autoinflammatory diseases.** PRAAS (Proteasome-Associated Autoinflammatory Syndrome): proteasome loss-of-function mutations decrease proteasome proteolytic activity and result in intracellular accumulation of polyubiquitylated proteins. These proteotoxic aggregates induce ER-stress which initiates the unfolded protein response (UPR). The IRE1 (inositol-requiring enzyme 1) arm of the UPR has been shown to contribute to the transcription of IFN-stimulated genes (ISG). A possible involvement of the integrated stress response (ISR) in this process is also discussed. NF-κB-mediated autoinflammation: PPR and cytokine receptor activation requires ubiquitylation for the induction of pro-inflammatory signaling. Depicted is the activation of the TNF receptor 1 (TNFR). The receptor-interacting protein kinase 1 (RIPK1) binds to the activated TFNR and is ubiquitylated with linear (M1-linked) poly-ubiquitin chains by the LUBAC complex or with K63-linked polyubiquitin. Polyubiquitylation is counterbalanced by the deubiquitinating enzymes (DUB) OTULIN and A20 in order to control the activation of the NF-κB and MAPK pro-inflammatory pathways. USP18 deficiency: USP18 besides its DUB activity also directly regulates IFN signaling. It is upregulated following different pro-inflammatory stimuli and directly inhibits JAK1, thereby acting as a negative feedback loop. Disruption of this negative feedback leads to overwhelming inflammatory IFN response.

**Figure 3 jcm-10-01783-f003:**
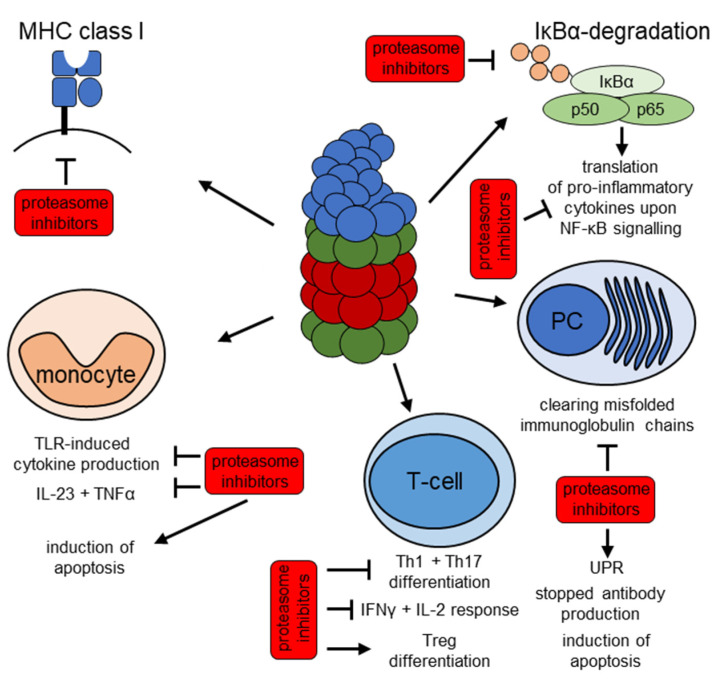
**Overview of the potential anti-inflammatory mechanisms of proteasome inhibitors.** Proteasome inhibitors including bortezomib, carfilzomib, ONX 0914 and KZR 616 were shown to exert anti-inflammatory effects via different mechanisms. These involve all cells: an influence on MHC class I antigen presentation (**top left**) and degradation of IκBα (**top right**), which results in NF-κB nuclear translocation and transcription of pro-inflammatory cytokines. Specific effects on cellular subsets include a targeted anti-inflammatory and proapoptotic effect on monocytes which results in a reduction in pro-inflammatory cytokine production upon TLR stimuli (**bottom left**). In T-cells, active IP are required for differentiation into Th1 and Th17 phenotypes (**bottom center**), whilst IP inhibition results in increased Treg differentiation and decreased IFN-γ production, as indicated. Plasma cells (PC) are especially sensitive to reduced proteasomal activity, which results in apoptosis most likely via activation of the UPR and in depletion of autoantibodies (**bottom right**).

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
