# Peer review of "Role of Proteasomes in Inflammation"

_jcm, 2021, doi:10.3390/jcm10081783_

Round 1

Reviewer 1 Report

Well written review, thorough searching of recent literature. I would suggest to remove the word chronic from the title.

Author Response

We thank the reviewer for the peer-review and agree that we do not limit our analysis in this manuscript to chronic inflammation and therefore followed his/her recommendation by changing the title into “Role of proteasomes in inflammation”

Reviewer 2 Report

Comment to the Authors
Title: Role of Proteasomes in chronic inflammation (jcm-1163436)
Journal: Journal of Clinical Medicine
Type: Review
Authors: Carl Christoph Goetzke et al
Carl Christoph Goetzk and Tilmann Kallinich (Contact)

The authors described proteasome-associated autoinflammatory syndromes (PRAAS: OMIM Nr. 256040).
They elegantly summarized the ubiquitin-proteasome-system, impaired its function, associated inflammatory diseases, and opportunity of therapeutic progress with excellent figures.
In their opinion, unravelling the pathogenesis of PRAAS will aid in better therapeutic approaches for PRAAS-patients.

Over all, this manuscript was well written. The findings obtained from the participants of this studies were very interesting. 
I hope their contributions to a scientific progress in this field.

Author Response

We thank the reviewer for the peer-review and also hope that we and/or others may soon be able to improve our knowledge of PRAAS pathogenesis.

Reviewer 3 Report

The authors expose in a broad and detailed way the role of the proteasome in the pathogenesis of autoinflammatory diseases. Taking the PRAAS as a model, they report the data available in the literature regarding the possible implication of UPR in the genesis of autoinflammation in other multifactorial polygenic-based diseases such as SLE, multiple sclerosis, Sjogren's syndrome and inflammatory intestinal diseases. This fascinating hypothesis opens the way to the potential use of drugs directed against this target in the treatment of rheumatological diseases. The authors report some data relating to the use of proteasome inhibitors in the treatment of SLE and other autoimmune diseases. It would be interesting to know if these drugs have been used in the treatment of PRAAS and if they have proved equally effective in this group of patients. 

Author Response

We thank the reviewer for reviewing our manuscript. We agree with him/her that, while proteasome inhibitors are well established, proteasome activators are not. Only very little data is available in this regard and, to the best of our knowledge, no small molecules specifically activating the proteasome have been brought to clinical trials so far. We are grateful to the referee for pointing this out and have clarified this point in the revised version of the manuscript (chapter 4 lines 453 – 460)